# PASRL: Stabilising Reinforcement Learning with Past Action-State Representation Learning

## Abstract

Although deep reinforcement learning (DRL) deals with sequential decision making problems, temporal information representation is absent from state-of-the-art actor-critic algorithms. The reliance on a single observation vector, representing information from only one time step, combined with densely connected neural networks, causes instability and oscillations in action smoothness. Therefore many applied DRL robotics control methods employ various reward shaping, low-pass filter and traditional controller-based methods to mitigate this effect. However, the interactions of these different parts hinders the performance of the original goal for the RL algorithm. In this paper we present a reinforcement learning algorithm extended with past action-state representation learning (PASRL), which allows for the end-to-end training of RL-based control methods without the need for common heuristics. PASRL is evaluated on the MuJoCo benchmark, showing smoother actions that preserve exploration, eliminate the need for extensive hyperparameter tuning, and provide a simple and efficient solution for enhancing action smoothness.

## 1 Introduction

Even though Reinforcement Learning (RL) Sutton & Barto (2018) is a powerful tool to deal with physical control problems, it exhibits a well-known instability regarding the smoothness of its predicted control actions Song et al. (2023)Mysore et al. (2021a). Oscillating, jerky control signals can degrade control performance and potentially damage the system Ibarz et al. (2021)Kim et al. (2022). This issue could be attributed to the reliance on a single observation vector, representing information only at time step $t$ and densely connected nature of the deployed neural network controller. Assuming that the state is a fully observable Markov Decision Process (MDP), instability could represent divergence in training, drops in performance across episodes, performance oscillations inside episodes and the actions taken by the agent could differ greatly from one time step to another. Furthermore, observation vectors that contain only the current time step's sensory recordings can lead to instabilities. This occurs when the agent lacks access to the complete observation space, transforming the underlying problem formulation into a Partially Observable Markov Decision Process (POMDP) Kaelbling et al. (1998). POMDPs could be induced by anomalies such as flickering, noise or data transmission loss in sensors during real-world applications. Or by the agent not having access to accurate information in its observation vector.

Stability issues in MDP formulated RL problems have been tackled by multiple methods and their mixtures (Figure. 1). Most commonly methods incorporate a motor behavior reward part that encourages improved action smoothness and the use of smaller action values into the desired reward function Liu et al. (2024). Others include past sensory readings and agent outputs into their observation vector, use the frame stacking of previous sensory observations Mnih et al. (2015), or employ state estimation models such as Kalman filters Kalman (1960), to better estimate the actual underlying state based on the sensory information received from the environment. Applied RL controllers commonly use low-pass filters to filter out large oscillations in the RL based control commands output and make use of traditional control algorithms such as PD or PID controllers Kaufmann et al. (2023)Luo et al. (2024)Reddy et al. (2018)Han et al. (2024)Jin et al. (2022).

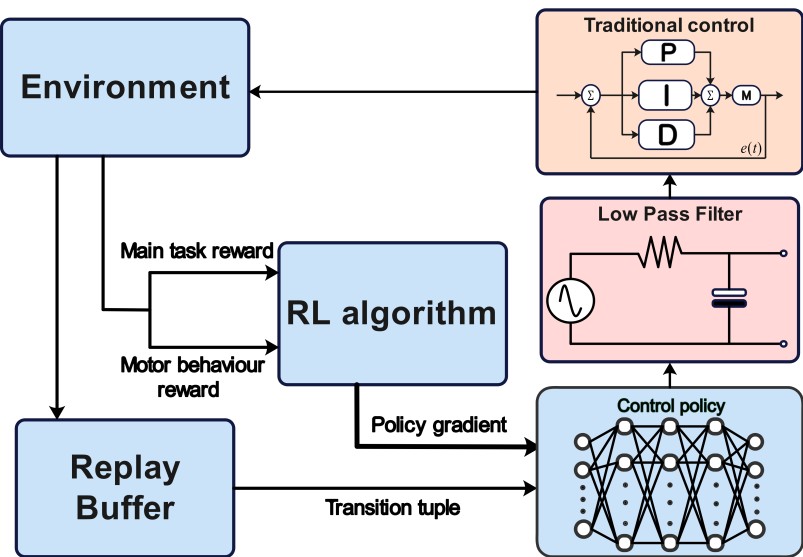

Figure 1: Applied reinforcement learning flowchart. In most applied reinforcement learning based control methods, RL based neural network controllers are augmented by a low-pass filter and/or a traditional control algorithm. The reward can be divided into two parts: the main reward task of the control and the behavior reward which forces the agent to output smooth actions and penalizes large actions.

However, what these methods do not take into account is that these modifications alter the original system's closed-loop dynamics leading to erratic control behavior Mysore et al. (2021b)Kim et al. (2022).

Densely connected layers have been the staple of most MDP formulated state-of-the-art algorithm Fujimoto et al. (2018)Haarnoja et al. (2018)Kuznetsov et al. (2020)Fujimoto et al. (2024). However, since these neural network structures contain no memory cells, they are prone to produce vastly different actions for concurrent time steps.

POMDP formulated reinforcement learning algorithms have been shown to mitigate these problems Dulac-Arnold et al. (2021) by either incorporating memory by stacking previous observations together ,Mnih et al. (2013) thus being able to turn partially observable MDPs to fully observable ones Hausknecht & Stone (2015) and mitigating the effect inaccurate state recordings could pose on the agent's observation vector. Furthermore, incorporating recurrent architectures within the agent enables the use of hidden states for memory integration. Additionally, it has been shown that recurrent architectures are effective even without frame-stacked observations Hausknecht & Stone (2015); Meng et al. (2021).

Recurrent neural network structure based agents have long been utilized in the Arcade Learning Environment (ALE) Bellemare et al. (2013). This environment offers interfaces with wide range of Atari 2600 games and has been a popular benchmark ever since. Most recurrent network-based agents rely on distributed training to avoid "representational drift," where stored hidden states generated by older network parameters differ significantly from those produced by the network at the current training step Kapturowski et al. (2018), Badia et al. (2020), Kapturowski et al. (2022), Espeholt et al. (2018)Horgan et al. (2018).

Improving the action smoothness generated by reinforcement learning agents has been explored via two main research lines. The modification of the RL training algorithms Shen et al. (2020), Mysore et al. (2021a), Chen et al. (2021), Yu et al. (2021), Kobayashi (2022), Zou et al. (2022) and by modifications of the policy network Takase et al. (2022), Song et al. (2023), Wang et al. (2024).

However, to the best of our knowledge no research has been conducted on recurrent reinforcement learning agents effect on action smoothness comparing against the training of reinforcement learning agents in concurrency with commonly used action smoothness reward, low-pass filter and traditional control heuristics. The training of this segmented system of RL controllers, low-pass filters and traditional control algorithms pose an issue since they are not optimized concurrently and rely on the correct guessing of various control parameters and cutoff frequencies. Furthermore the incorporation of an action smoothness term causes the agent to maximize the balance between achieving the main objective and minimizing abrupt changes in actions, instead of solely maximizing the primary goal.

It is also important to note that taking smooth actions is not always optimal. Overly smoothed actions can restrict the agent's exploration, potentially limiting its performance and preventing it from fully exploring the state-action space needed to achieve an optimal policy. Moreover, some environments demand rapid action responses where swift or highly reactive control strategies are ideal. Hence, an effective algorithm should balance responsiveness to accommodate rapid changes while minimizing unnecessary oscillations in action selection.

In this paper we propose a non-distributed recurrent reinforcement learning agent, with learned hidden states. Our approach can be thought of as an extension to TD7 Fujimoto et al. (2024), which also learns decouples state and state-action embeddings. We augment this already existing pipeline by creating time-dependent embeddings. The proposed RL agent could be trained end-to-end without the commonly employed heuristics present in applied reinforcement learning methods. We evaluate this algorithm's performance in two metrics: the control methods achieved reward in the main task of the environment and the action smoothness of the created control strategy. Our findings show that recurrent reinforcement learning agents achieve comparable task performance to mixed traditional and RL-based controllers, while generating substantially smoother actions without relying on heuristics or compromising exploration.

## 2 BACKGROUND

Reinforcement learning formulates problems as a Markov Decision Process Bellman (1957)Sutton & Barto (2018). An MDP can be described as a tuple of 5 $(S, A, R, p, \gamma)$, containing $S$ the state space, $A$ action space, $R$ reward function, $p$ dynamics model and discount factor $\gamma$. In RL the objective is to find an optimal policy $\pi_\theta : S \to A$, that maps state $s \in S$ to an action $a \in A$, in a way that maximizes the discounted accumulative reward $\sum_{t=1}^{\infty} \gamma^{t-1} \cdot r_t$, with parameters $\theta$.

Recurrent Neural Networks (RNNs) are widely applied in reinforcement learning (RL) to address tasks involving temporal dependencies, where decisions depend not only on the current observation but also on past experiences. By maintaining a hidden state that evolves over time, RNNs enable RL agents to incorporate historical information, making them particularly effective in partially observable environments, such as Partially Observable Markov Decision Processes (POMDPs). The most commonly used RNN variants in RL are Long Short-Term Memory (LSTM) networks Hochreiter & Schmidhuber (1997) and Gated Recurrent Units (GRUs) Cho (2014), both of which mitigate the vanishing gradient problem and improve performance in tasks that require memory and sequential decision-making.

State-Action Learning Embeddings (SALE) Fujimoto et al. (2024) are designed to improve RL algorithms by effectively capturing observation space structure and transition dynamics. It employs two encoders: $f$ transforms the state $s$ into an embedding $z_s$ and $g$ combines $z_s$ with action $a$ to create a state-action embedding $z_{sa}$. SALE serves as the principle component for TD7 Fujimoto et al. (2024), which is an improved version of TD3 Fujimoto et al. (2018).

## 3 ACTION SMOOTHNESS ISSUE OF CURRENT STATE-OF-THE ART METHODS

Although recurrent neural networks (RNNs) are widely used in POMDP tasks, the broader research community has not fully adopted them into MDP tasks. Instead, many deep reinforcement learning (DRL) algorithms are typically used off-the-shelf relying on densely connected neural networks that lack memory of previous inputs or outputs. This is problematic, as reinforcement learning is fundamentally aimed at addressing sequential decision-making challenges.

We show that the sole reliance on feed forward densely connected networks introduces instabilities present in the action smoothness of a trained agents performance. Further evidence of RL instabilities is provided in the Appendix.

### 3.1 INSTABILITY IN ACTION OUTPUTS

Stability/smoothness of the action outputs is not commonly explored in reinforcement learning algorithmic papers. Although not relevant to the usual return score representation commonly found in papers, it provides valuable insight into the feasibility of the learned control strategy in many real world applications.

To investigate how attainable the learned action outputs are, we can evaluate the rate of change of the rate of change of the outputs, which we approximate by using the second derivative of the actions.

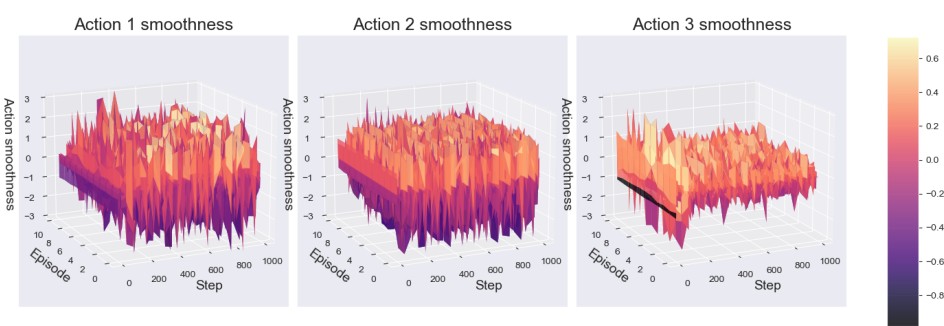

Figure 2: The action smoothness of a TD7-based agent trained for 3M time steps in the Hopper environment. The results were evaluated across ten episodes.

The agent's lack of memory is evident upon examining the output values. (Figure. 2) Each output at a given time step has no connection to the output before or after it. Also the magnitude of the second derivative of these output poses a real difficulty in achieving these output values in real-world scenarios. Although periodic oscillations are expected, because of the nature of the environment any structured change in the smoothness of the actions is absent.

## 4 METHODOLOGY

In this section we introduce our past action-state information learning method, as well as perform in-depth empirical evaluations for the design choices when using past information augmentation. PASRL is built on TD7 with additional recurrent encoder structure, with prioritized recent experience replay to alleviate recurrent state staleness Kapturowski et al. (2018) and use pink noise Eberhard et al. (2023) for added exploration benefits with more correlated noise.

### 4.1 PAST ACTION-STATE REPRESENTATION LEARNING

The aim of past action-state representation learning is to learn time dependent embeddings $(z_t^{sa}, z_t^s)$, which is able to capture the time dependent change of the observation space and environment characteristics. PASRL augments the encoder pair $(f, g)$ present in TD7, with a recurrent bottleneck hidden layer, with learnable hidden states. In PASRL $f(\tilde{s}, h_t)$ encodes information from state $\tilde{s}$ and the hidden state of the state encoder $h_t$ into time dependent state embedding $z_t^s$ and $g(z^s, h_t, \tilde{a})$ encodes state $\tilde{s}$ and action $\tilde{a}$ into a time dependent state action embedding $z_t^{sa}$.

### 4.2 REPLAY BUFFER MODIFICATIONS

In order to train our algorithm with hidden states of the recurrent layers present, we modify our replay buffer to store transition tuples as well as the current $h_t$ and next hidden states $nh_t$ at a given

time step $(\tilde{s}, \tilde{a}, \tilde{s}', r, h_t, nh_t)$. We also enhance the state and action vectors by defining them as fixed-length sequences of history ($h_l = 10$), referred to as $\tilde{s}$ and $\tilde{a}$. These sequences are updated by appending the current time step observation $s$, and action $a$ at the end for $\tilde{s}$ and $\tilde{a}$ and removing the oldest information. These observation and action vectors are initialized by zeroes and stored without crossing episode boundaries.

To alleviate recurrent state staleness and representation drift we employ a prioritizing recent experience replay sampling method Wang & Ross (2019) to prioritize the probability of sampling transition tuples created by more up-to-date network parameters.

Therefore in a given update phase we make $K$ mini-batch updates. Suppose $N$ is the size of the replay buffer, then for the $k$th update, where $1 \leq k \leq K$, we perform uniform sampling from the most recently stored $c_k$ data points, which is defined by

$$c_k = max\{N \cdot \eta^{\frac{k \cdot m}{K}}, c_{min}\} \tag{1}$$

in which equation $\eta$ is the hyperparemeter determining how much prioritization is assigned to newer samples, $c_{min}$ determines the minimum sub-buffer range from which we can sample, and $m$ is an environment-dependent variable that is the maximum steps inside an episode.

### 4.3 RECURRENT LAYER MODIFICATION

For the recurrent layer type in our algorithm we selected GRU layers, since compared to LSTMs they have less parameters per unit, therefore we can increase the bottleneck size with less parameters. Furthermore, GRU layer based methods typically achieve the best performance in POMDP based benchmarks Morad et al. (2023).

To make the comparisons fair and to keep the encoder's parameter size from growing substantially due to more parameters found in a GRU unit, compared to DNN units, we have chosen to reduce the number of layers our encoders utilize, to ensure the encoder is capable of creating meaningful embeddings to the actor and critic networks. Therefore not hindering the overall training process by outputting not trained embeddings.

To ensure that the encoders utilize all the recurrent bottleneck layer's neurons, we used a dropout Srivastava et al. (2014) of 20%.

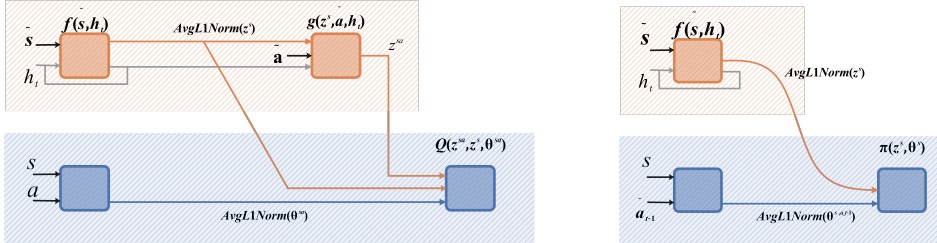

(a) Modified value function $Q$ of PASRL.    (b) Modified policy function $\pi$ of PASRL.

Figure 3: The flow chart of the information propagation in PASRL. PASRL builds on the encoder structure used in TD7 augmenting it with the use of recurrent layered encoders with the propagation of hidden states $h_t$ between the encoders to allow for the encoding of time dependencies inside an episode. (Figure inspired by Fujimoto et al. (2024).)

### 4.4 NOISE FOR EXPLORATION

Actor-critic reinforcement algorithms typically encourage exploration via adding noise to the output actions or by target policy noise. These methods utilize white noise for both exploration methods. However, it has been shown that white noise is not able to sufficiently explore action spaces, and the

use of a more correlated color noise. for example pink noise, achieves better results for the agent Eberhard et al. (2023).

In our method we utilize pink noise for the exploration during actions chosen during training and white noise exploration for the target policy noise. We further utilize the addition of white noise to the sampled next hidden states during the training of the critic networks.

### 4.5 EFFECTS OF REPRESENTATIONAL DRIFT

For the training of recurrent reinforcement learning models two methods are commonly described Hausknecht & Stone (2015). The first replays entire episode trajectories, while the second utilizes the common sampling paradigm for training. Although these methods follow different chains of thought, they overall lead to the same performance, therefore in PASRL, we utilize the common sampling paradigm found in TD7.

As for the use of the hidden states values we can also divide them into two categories.

1. Zeroing out the hidden states of sampled transition tuples

2. Storing the hidden states of the transition tuples.

The first approach appeals in its simplicity for implementation, however limits the networks temporal information modeling capability. While the second suffers from an effect called representational drift, where the stored hidden states generated by a sufficiently old network parameters causes discrepancy, since the updated network's parameter generated hidden states do not align with the stored ones.

In order to measure recurrent state staleness and representational drift, we can measure the Q-value discrepancy Kapturowski et al. (2018) between Q-values generated by the network's up-to-date hidden states versus the stored hidden states.

$$\Delta Q = \frac{\|q_t(\hat{h}_t, \hat{\theta}) - q_t(h_t, \hat{\theta})\|_2}{|max(q_t(\hat{h}_t, \hat{\theta}))|} \tag{2}$$

Where $\hat{h}_t$ are the hidden states generated by the up-to-date state encoder network parameters, and $h_t$ are the stored hidden states generated by the encoder during a point of previous training. With $\hat{\theta}$ denoting the current parameters of the network.

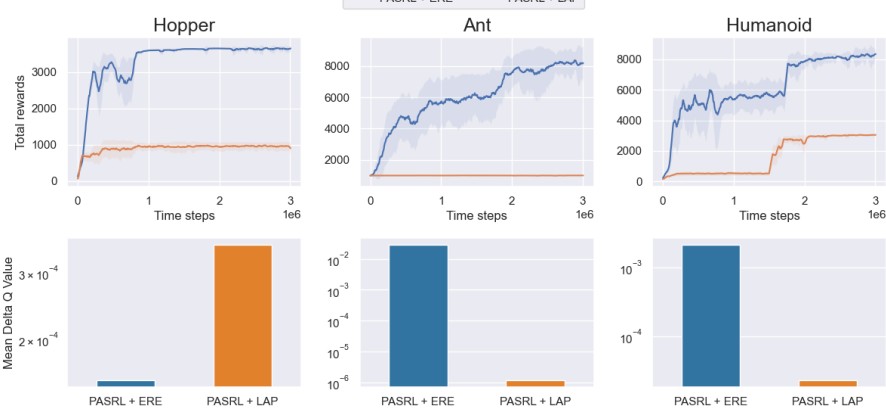

Figure 4: Delta Q discrepancy as a measure for representational drift. The results were achieved in 3 MuJoCo environments over 3 seed, with delta Q values being stored between 1M and 3M time steps. The shaded area captures the standard deviation of the average performance.

In Figure 4 we show that emphasizing recent experience replay (ERE) Wang & Ross (2019) provides an antidote for counteracting representational drift as we compare it to the prioritized experience replay Fujimoto et al. (2020) present in TD7.

PASRL with ERE is able to overcome representational drift, meanwhile PASRL with LAP fails to learn any meaningful policies, leading to minimal $\Delta Q$ values. PASRL with ERE is capable of solving this issue and effectively minimizes $\Delta Q$ difference between the hidden variables generated by the encoder network's weights at the current training step and the stored $h_t$ values.

## 5 RESULTS

In this section, we evaluate the main task reward, which refers to the original reward without any motor control penalties, and the action smoothness of the control policy based on PASRL, comparing it against TD7 across various commonly constructed applied reinforcement learning control loops. These include PASRL without any modifications, TD7 with an additional action smoothness reward component (TD7 + AC), TD7 with both the action smoothness reward and a low-pass filter (TD7 + AC +LPF), TD7 with the action smoothness reward and a PD controller (TD7 + AC + PD), and finally, TD7 with the action smoothness reward, a low-pass filter, and a PD controller (TD7 + AC + LPF + PD).

We obtain these results using 4 different OpenAI gym Brockman (2016) MuJoCo Todorov et al. (2012) environments. A detailed description of the used hyperparameters, baselines and experimental setup is included in the Appendix.

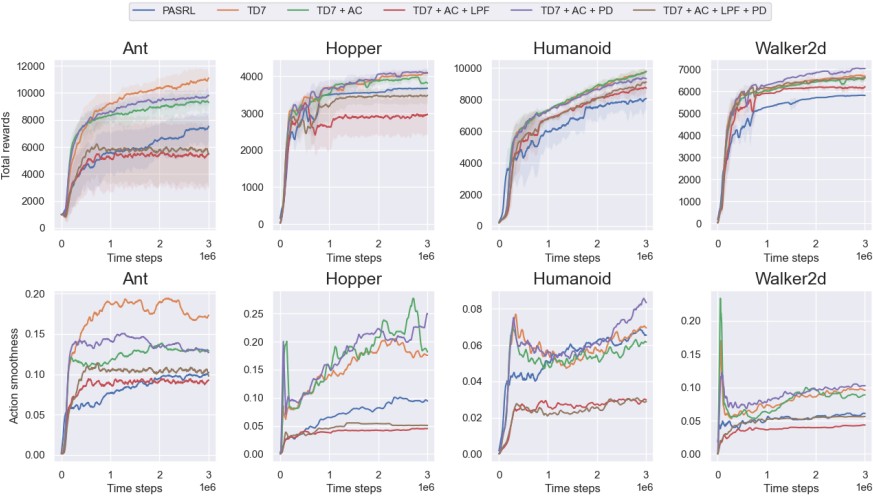

Figure 5: Learning curves on the MuJoCo benchmark. Results are averaged over 10 seeds, except in the case of the Walker2d environment, where only 7 seeds were used. The shaded area captures the standard deviation (std), around the average performance.

Figure 5 presents the learning curves for the different control strategies. Table 1 highlights the quantitative results, summarizing the performance during and at the end of training. The quantitative results for the action smoothness values are provided in Table 2. The learning curves indicate that, as the reward of TD7-based agents increases exponentially in the early stages of training, there is a significant spike in action smoothness. We attribute this to the exploration phase, where the agent has not yet developed an optimal policy.

Additionally, we observe that applying action smoothing through a low-pass filter (LPF) in mixed control methods impedes the agent's convergence in some cases, as it constrains exploration. This negative impact is more pronounced in environments with larger action spaces, such as Ant, compared to smaller action space environments like Hopper. The Humanoid and Walker2d environments are an exception to this pattern, which have similar tasks. Also in the first case this could be due to the limited range of actions available to the agent.

| Environment | Time Step | PASRL | TD7 | TD7+AC | TD7+AC+LPF | TD7+AC+PD | TD7+AC+LPF+PD |
|---|---|---|---|---|---|---|---|
| Ant | 300k | 3750 ± 1785 | 6222 ± 1442 | 6907 ± 769 | 3775 ± 2080 | 6818 ± 803 | 3830 ± 2141 |
| | 1M | 5596 ± 1498 | 9276 ± 577 | 8314 ± 402 | 4941 ± 2453 | 8858 ± 217 | 5776 ± 2380 |
| | 3M | 7676 ± 898 | 11230 ± 67 | 9234 ± 1141 | 5564 ± 2561 | 9749 ± 973 | 5260 ± 2707 |
| Hopper | 300k | 2313 ± 311 | 3146 ± 106 | 3032 ± 200 | 2791 ± 339 | 3260 ± 126 | 2872 ± 311 |
| | 1M | 3500 ± 5 | 3692 ± 49 | 3502 ± 213 | 2606 ± 613 | 3583 ± 420 | 3109 ± 508 |
| | 3M | 3686 ± 5 | 4096 ± 73 | 3810 ± 286 | 2981 ± 468 | 4074 ± 192 | 3508 ± 178 |
| Humanoid | 300k | 4200 ± 1583 | 5358 ± 1000 | 5391 ± 1097 | 4060 ± 1169 | 5228 ± 938 | 4252 ± 1305 |
| | 1M | 5936 ± 889 | 7259 ± 292 | 7409 ± 22 | 6879 ± 23 | 7305 ± 16 | 6764 ± 156 |
| | 3M | 8090 ± 652 | 9768 ± 220 | 9813 ± 22 | 8723 ± 451 | 9335 ± 615 | 9169 ± 20 |
| Walker2d | 300k | 4069 ± 413 | 5275 ± 358 | 5104 ± 324 | 4991 ± 451 | 4729 ± 842 | 5250 ± 507 |
| | 1M | 5296 ± 12 | 6086 ± 18 | 6029 ± 20 | 5912 ± 34 | 6288 ± 100 | 6164 ± 70 |
| | 3M | 5824 ± 14 | 6748 ± 208 | 6627 ± 30 | 6234 ± 44 | 7041 ± 15 | 6652 ± 72 |

Table 1: Average reward performance on the selected MuJoCo benchmark at 300k, 1M, and 3M time steps. ± captures the standard deviation of the averaged main task rewards.

| Environment | Time Step | PASRL | TD7 | TD7+AC | TD7+AC+LPF | TD7+AC+PD | TD7+AC+LPF+PD |
|---|---|---|---|---|---|---|---|
| Ant | 300k | 0.0627 | 0.1433 | 0.1176 | 0.0739 | 0.1456 | 0.0779 |
| | 1M | 0.0764 | 0.1877 | 0.1170 | 0.0827 | 0.1430 | 0.1029 |
| | 3M | 0.1045 | 0.1754 | 0.1281 | 0.0932 | 0.1260 | 0.0945 |
| Hopper | 300k | 0.0257 | 0.0788 | 0.0810 | 0.0301 | 0.0970 | 0.0338 |
| | 1M | 0.0650 | 0.1436 | 0.1599 | 0.0376 | 0.1506 | 0.0435 |
| | 3M | 0.0938 | 0.1766 | 0.1794 | 0.0454 | 0.2489 | 0.0511 |
| Humanoid | 300k | 0.0426 | 0.0702 | 0.0699 | 0.0214 | 0.0759 | 0.0242 |
| | 1M | 0.0497 | 0.0572 | 0.0488 | 0.0263 | 0.0571 | 0.0211 |
| | 3M | 0.0656 | 0.0690 | 0.0612 | 0.0298 | 0.0832 | 0.0290 |
| Walker2d | 300k | 0.0403 | 0.0551 | 0.0554 | 0.0312 | 0.0696 | 0.0416 |
| | 1M | 0.0564 | 0.0748 | 0.0618 | 0.0364 | 0.0788 | 0.0524 |
| | 3M | 0.0610 | 0.0957 | 0.0883 | 0.0433 | 0.1024 | 0.0564 |

Table 2: Average action smoothness values on the selected MuJoCo benchmark at 300k, 1M, and 3M time steps.

Integrating traditional control heuristics with a RL agent in these settings adds complexity and demands extensive hyperparameter tuning. Often this results in solutions that struggle to improve action smoothness consistently across diverse environments. PASRL strikes a balance between the high performance of top mixture methods and the action smoothness of LPF-based approaches. It avoids the initial spike in action smoothness seen during early training and does not over-smooth actions to the detriment of exploration, as demonstrated in the Humanoid environment results.

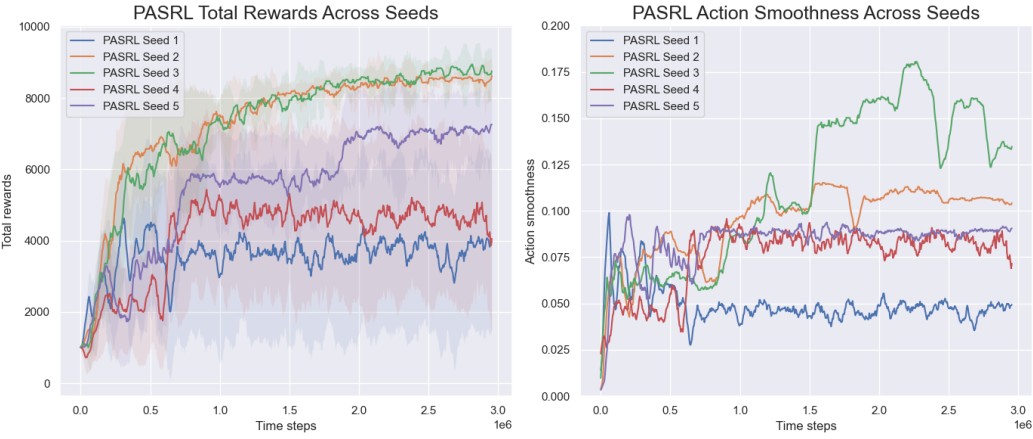

Figure 6: PASRL performance across different seeds. The results are achieved over the AntSchulman et al. (2015) MuJoCo environment. The shaded area captures the standard deviation of the evaluation episodes.

Figure 6 illustrates the performance variability of PASRL across different seeds, showing that it can rival the top reward performance of the best TD7 control mixture method. Notably, PASRL achieves smooth action outputs without relying on hand-tuned components like low-pass filters or

PD controllers for action regularization, which are commonly used in TD7-based methods. This capability demonstrates PASRL's potential to simplify the control process by eliminating the need for auxiliary reward shaping or extensive parameter tuning for outside the primary RL framework. By eliminating the need for extensive expertise in designing controllers where smoothness is critical, PASRL simplifies the process of achieving effective control. It removes the complexity associated with tuning traditional control methods to the specific characteristics of the environment.

## 6 CONCLUSION

The action smoothness of state-of-the-art RL algorithms is often addressed by adding smoothness reward terms, low-pass filters, or traditional control methods. However, these approaches can hinder performance. For this reason we introduce PASRL, a method to learn time-dependent state-action embeddings to create smoother action controls.

This paper highlights the issue of action smoothness found in RL algorithms using densely connected networks, and show how this could be overcome without changing the mathematical models behind the algorithm's training. We also incorporate various advances in exploration and optimizers.

PASRL is able to generate smooth actions without requiring manually designed reward parts or additional controllers, such as low-pass filters or PD controllers. This reduces the complexity of parameter tuning in applied RL cases. As a general-purpose technique, PASRL offers an alternative for reinforcement learning tasks where the smoothness of the controller is amongst key priorities. We found that PASRL is even able to match the performance of commonly employed RL and traditional control methods in some environments, while outputting smoother actions.

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

# A APPENDIX

## A.1 HYPERPARAMETERS

The action space for the environments is in the range of [-1, 1]. Most hyperparameters match TD7.

Our algorithm differs from TD7 in the following changes:

1. The use of hidden states and GRU neurons inside the encoders

2. The use of NAdam instead of Adam

3. The use of a different exploration strategy, using Pink noise for exploration during episodes and adding white noise to the next hidden noise values $hnt$ during the training of the target value function $Q_t$

4. We furthermore decided to leave out warm-up period for the sake of learning the hidden variable transitions of the agent from the start of the training.

## A.2 ALGORITHM DESCRIPTIONS

Pseudocode for PASRL is described in Algorithm 1.

| | Hyperparameter | Value |
|---|---|---|
| TD3 Fujimoto et al. (2018) | Target Policy noise $\sigma$ | $N(0, 0.2^2)$ |
| | Target Policy noise clipping $c$ | (-0.5, 0.5) |
| | Policy update frequency | 2 |
| ERE Wang & Ross (2019) | $c_{min}$ | 25k |
| | $\eta$ | 0.994 |
| TD3 + BC Fujimoto & Gu (2021) | Behavior cloning weight $\lambda$ (Online) | 0.0 |
| | Behavior cloning weight $\lambda$ (Offline) | 0.1 |
| Policy Checkpoints | Checkpoint criteria | minimum |
| | Early assessment episodes | 1 |
| | Early time steps | 20 |
| | Early time steps | 750k |
| | Criteria reset weight | 0.9 |
| PASRL | GRU neurons | 80 |
| | GRU layers | 2 |
| Exploration | Initial random exploration time steps | 0 |
| | Color parameter beta | 1.0 |
| | Noise scale | 0.3 |
| | Target Policy hidden noise | $N(0, 0.1)$ |
| Common | Discount factor | 0.99 |
| | Replay buffer capacity | 1M |
| | Mini-batch size | 256 |
| | Target update frequency | 250 |
| Optimizer | (Shared) Optimizer | NAdam Dozat (2016) |
| | (Shared) Learning rate | 3e-4 |

Table 3: PASRL Hyperparameters

---

**Algorithm 1** Online PASRL

---

1: **Initialize:**
2: Policy $\pi_{t+1}$, value function $Q_{t+1}$, encoders $(f_{t+1}, g_{t+1})$.
3: Target policy $\pi_t$, target value function $Q_t$, fixed encoders $(f_t, g_t)$, target fixed encoders $(f_{t-1}, g_{t-1})$.
4: Checkpoint policy $\pi_c$, checkpoint encoder $f_c$.
5: **for** Every episode **do**
6:     Reset hidden states $h_t$ of the encoder
7:     Generate episodic pink action noise
8:     **if** checkpoint condition **then**
9:         **if** actor $\pi_{t+1}$ outperformes checkpoint policy $\pi_c$ **then**
10:             Update checkpoint networks $\pi_c \leftarrow \pi_{t+1}, f_c \leftarrow f_t$.
11:         **end if**
12:     **end if**
13:     **for** episode in episodes_since_training **do**
14:         **for** time step: t=1,..,K in episode **do**
15:             calculate $c_k$
16:             Sample transitions from ERE replay buffer, based on $c_k$
17:             Train encoder, value function, and policy (accordingly to Fujimoto et al. (2024))
18:             **if** passed steps > target_update_frequency **then**
19:                 Update tartget networks (based on Fujimoto et al. (2024)
20:             **end if**
21:         **end for**
22:     **end for**
23: **end for**

---

### A.3 BASELINES HYPERPARAMETERS

### A.3.1 TD7

Our TD7 implementation uses the exact hyperparameters as described by the author at `https://github.com/sfujim/TD7`.

| | Hyperparameter | Value |
|---|---|---|
| TD3 Fujimoto et al. (2018) | Target Policy noise $\sigma$ | $N(0, 0.2^2)$ |
| | Target Policy noise clipping $c$ | (-0.5, 0.5) |
| | Policy update frequency | 2 |
| LAP Fujimoto et al. (2020) | Probability smoothing $\alpha$ | 0.4 |
| | Minimum priority | 1 |
| TD3 + BC Fujimoto & Gu (2021) | Behavior cloning weight $\lambda$ (Online) | 0.0 |
| | Behavior cloning weight $\lambda$ (Offline) | 0.1 |
| Policy Checkpoints | Checkpoint criteria | minimum |
| | Early assessment episodes | 1 |
| | Early time steps | 20 |
| | Early time steps | 750k |
| | Criteria reset weight | 0.9 |
| Exploration | Initial random exploration time steps | 25k |
| | Exploration noise | $N(0, 0.1^2)$ |
| Common | Discount factor | 0.99 |
| | Replay buffer capacity | 1M |
| | Mini-batch size | 256 |
| | Target update frequency | 250 |
| Optimizer | (Shared) Optimizer | Adam Kingma & Ba (2014) |
| | (Shared) Learning rate | 3e-4 |

Table 4: TD7 Hyperparameters

### A.3.2 ACTION SMOOTHNESS

The smoothness of an action policy can be approximated by the second derivative of the actions.

$$\text{Action Smoothness} = \frac{1}{N} \sum_{t=1}^{N} (a_{t+2} - 2a_{t+1} + a_t)^2 \tag{3}$$

Our action smoothness metrics for the baseline methods and PASRL were calculated by Equation 3.

The action smoothness reward part enhanced environment for Ant has the same equation for the calculation of action smoothness, with a weight of $w_{as} = 0.1$.

### A.3.3 LOW-PASS-FILTER

Low-pass filters in reinforcement learning controllers aim to smooth out high-frequency fluctuations in the actions generated by the policy. To implement a low-pass filter in our reinforcement learning (RL) controller, two key parameters must be defined: the sampling frequency $f_s$ and the cutoff frequency $\omega_c$. The sampling frequency determines the time step, calculated as $\Delta t = 1/f_s$, and is typically provided by the reinforcement learning environment. The cutoff frequency defines the filter's response characteristics. In our implementation, we selected a cutoff frequency of $20Hz$ to effectively balance responsiveness and smoothness of the policy.

In our low-pass-filter implementation we utilize a first-order Infinite Impulse Response (IIR) filter, to minimize the delay in the control policy, which can be described by:

$$y_t = b_0 \cdot \mathbf{a_t} + b_1 \cdot \mathbf{a_{t-1}} + a_1 \cdot \mathbf{y_{t-1}} \tag{4}$$

Where the coefficients in Equation 4 $b_0$ and $b_1$ control the contribution of the current and previous inputs, respectively, while $a_1$ determines how much of the previous output affects the current output.

The variable $a_t$ represents the current input action, which is the new action generated by the reinforcement learning policy. The term $a_{t-1}$ denotes the previous action, allowing the filter to take into account the action taken in the prior time step. Finally, $y_{t-1}$ represents the previous filtered action output, which serves as feedback to the current output. At the end the final filtered action output $y_t$ is used as the input for the environment.

### A.3.4 PD CONTROLLER

A Proportional-Derivative (PD) controller is a widely used traditional control paradigm in various robotic applications used to enhance the performance of action outputs. The PD controller combines two components: the proportional $P$ term, which provides an immediate response to the current error, and the derivative $D$ term, which predicts future error based on its rate of change.

A PD controller can be described by the following formula:

$$u(t) = K_p \cdot e(t) + K_d \cdot \frac{de(t)}{dt} \tag{5}$$

In which equation: $u(t)$ is the control output, $e(t)$ is the error signal (in our case the policy action), $K_p$ is the proportional gain parameter, and $K_d$ is the derivative gain.

In the context of reinforcement learning, the PD controller is employed to smooth the action outputs of the policy. The proportional component reacts directly to the error in the action, adjusting the control output to reduce this error. The derivative component, on the other hand, aims to mitigate oscillations in the agent's output actions. In our implementation we have utilized a PD controller with proportional gain $K_p = 1.0$ and derivative gain $K_d = 0.05$ to provide a rapid response to the given actions generated by the policy.

### A.4 FURTHER INSTABILITIES IN RL

### A.4.1 INSTABILITY ACROSS EPISODES

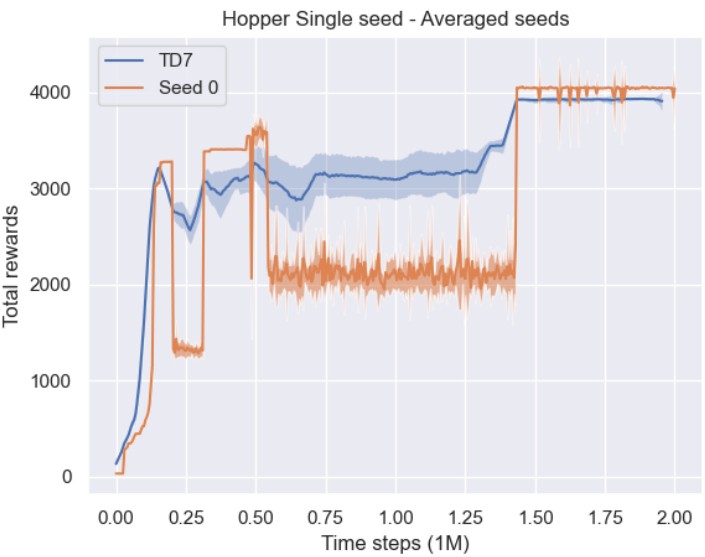

Figure 7: Difference in curve smoothness for TD7. Averaged vs single seed representation of the results. Averaged over 5 different seeds over 2M time steps.The shaded area in the TD7 algorithm represents rewards averaged over 10 evaluation episodes per seed, and then averaged across all seeds. For Seed 0, the shaded region specifically reflects the standard deviation among the evaluation episodes within that seed.

The fair comparison of RL algorithms have long been a challenge in the evaluation of these algorithm's results. The issue stems from the variance of the stochastic environment and in the learning initialization Henderson et al. (2018). With different seeds having drastically different performances across different seeds.

To counteract this issue most algorithms report their performance according to the following standard: They evaluate every $N_{freq}$ steps, over multiple evaluation episodes $N_{ep}$ over a number of seeds $N_{seeds}$ and finally they smooth the plot by averaging over a given window size $N_{window}$ Fujimoto et al. (2024).

Although this standard procedure makes the plots easier to understand and allows for fair comparison across algorithms, it creates a false sense of stability regarding RL algorithms. (Fig.7)

Where even though the papers report smooth learning curves single seed runs could yield a much different result during training.

### A.4.2 INSTABILITY IN EPISODES

A common performance reporting standard for RL algorithms has long been the plotting of the episodic return along a given time step of training. Although it gives a clear picture for performance throughout the episode it is a surface level reporting since the intricacies of how it achieved these rewards inside the episode is left unanswered.

When examining the rewards the agent accumulates throughout the episode we can make the following observations. (Fig. 8)

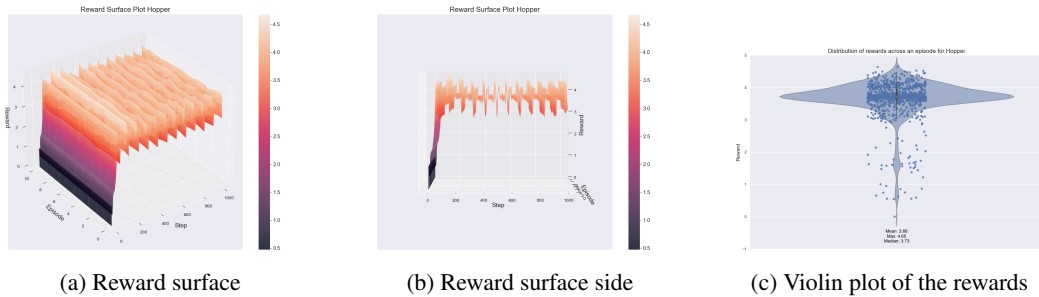

(a) Reward surface        (b) Reward surface side        (c) Violin plot of the rewards

Figure 8: In episode performance fluctuations present in TD7. Subfigure a, shows the reward surface of a trained TD7 agent on the Hopper environment across 1000 time steps and 10 different episodes. Subfigure b, shows the reward surface from the side to further showcase the oscillations during time steps. Subfigure c, shows the violin plot of the accumulated rewards across the mentioned 1000 time steps and 10 different episodes.

Firstly is that the rewards at the beginning of the episode vastly under perform compared to rewards in the middle or at the end of the training. (Fig8a.) Secondly the rewards fluctuate similar to a sinusoid function during inference. (Fig 8b.) Finally the performance can exhibit significant fluctuations around the mean and in some cases it could deviate substantially from the mean reward of the episode. (Fig 8c.)

### A.5 EVALUATING DESIGN CHOICES

### A.5.1 OPTIMIZER

Although Actor-Critic algorithms have been around for ten years most commonly used state of the art algorithms use Adam Kingma & Ba (2014). While Adam is a trusted and well performing neural network optimizer, newer versions of this optimizer mainly NAdam Dozat (2016) and ND-Adam Zhang et al. (2017) have shown that they could improve on its performance. In this section we discuss and evaluate their performance in a reinforcement learning setting.

NAdam (Nesterov-accelerated Adaptive Moment Estimation) is an enhanced version of the Adam optimizer, combining the advantages of the Adam and Nesterov momentum methods. The Nadam

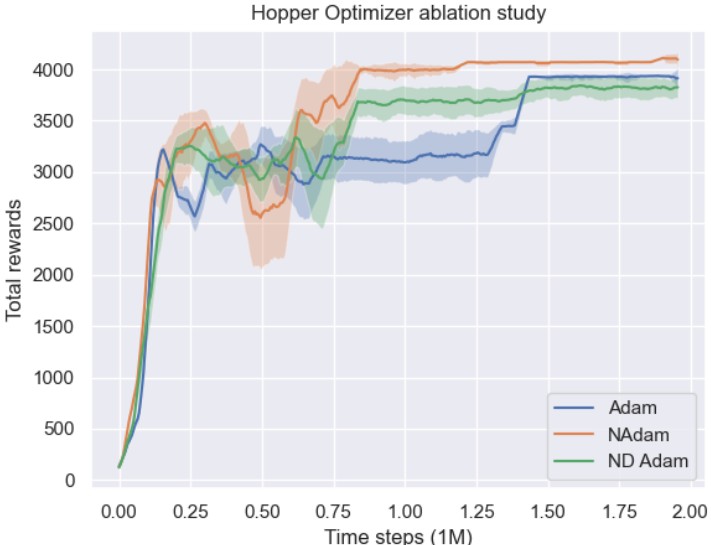

Figure 9: Optimizer Ablation study. Learning curves on the Hopper MuJoCo benchmark. Results are averaged over 5 seeds. The shaded area captures the standard deviation of the evaluated episodes at a given time step.

algorithm merges the adaptive learning rate of Adam with the Nesterov-accelerated gradient method, achieving faster and more stable convergence, especially in deep neural networks.

ND-Adam (Normalized Direction-preserving Adam) is an improved optimization algorithm, its primary aim is to enhance the learning process's efficiency by maintaining the gradient's direction while normalizing it.

In this paper we provided an ablation study where we compared the performance of Adam, NAdam and ND-Adam Fig. 9. From which we can observe that NAdam has a slight performance boost compared to other methods. For this reason we used NAdam optimizer for PASRL.

A.6   EXPERIMENTAL DETAILS FOR REPRODUCIBILITY

All experiments were conducted using fixed seeds for Gymnasium, Torch Paszke et al. (2019), and Numpy Harris et al. (2020) to ensure consistency. The results were evaluated in evaluation mode (without exploration noise). For evaluation, we used 10 seeds ranging from 0 to 9, except in the case of the Walker2d environment, where only 7 seeds were used. All baselines and our method were evaluated on the same fixed seeds 0-9. The evaluation was performed every 5000 time steps, with checkpointing enabled for TD7. The results were averaged over the 10 evaluation episodes inside a seed and then were averaged across seeds. Windowing smoothing with a window size of 10 was applied to display the results.

The action smoothness results were computed using the second-order derivative of each action output, taking the absolute value, and then averaging the smoothness values across evaluation episodes. These averaged values were further averaged across all seeds to obtain the final smoothness measure.

A.7   LIMITATIONS

It is important to note that while PASRL aims to enhance action smoothness in applied RL controllers, it is not without limitations. Due to the black-box nature of neural networks, the safety of selected actions remains a concern, and implementing general fail-safe mechanisms is essential to address this issue.

