# OpenReview forum: "PASRL: Stabilising Reinforcement Learning with Past Action-State Representation Learning"
_ICLR.cc/2025/Conference — Submitted to ICLR 2025_

### Official Review · Reviewer_sexv · 2024-10-27

**Soundness:** 1
**Presentation:** 1
**Contribution:** 2
**Rating:** 3
**Confidence:** 4

**Summary:**

This paper proposes a new DRL method called PASRL based on TD7 method, extended with past action-state representation learning. PASRL can achieve smoother action sequences compared to existing DRL methods through end-to-end training without heuristics design. The experiment on the Ant control task demonstrates the effectiveness of the new method.

**Strengths:**

1. Improving action smoothness is a meaningful research topic in DRL domains, especially in continuous control tasks such as robotics control and auto driving.
2. The key idea of the new method PASRL is easy to follow and implement.

**Weaknesses:**

1. More related works need to be introduced in this paper. Improving action smoothness is the key contribution of this paper, which has been widely researched recently in DRL domain. The following are some related works:

   1. LipsNet[1]: reduce action fluctuation of DRL methods in continuous control tasks through constraining the Lipschitz constant of policy networks utilizing Multi-dimensional Gradient Normalization (MGN).
   2. TAAC[2]:  introduce a switch policy to make binary choices on whether to use the current new action or the last action at each step in continuous control tasks.
   3. PIC[3]: introduce action inertia into discrete DRL methods, where the policy tends to take the same actions at the previous step.

   Please find more related works and give a more detailed description of this research filed in the manuscript.

2. The experiment is insufficient and cannot demonstrate the effectiveness of the new method:

   1. The experiment result of PASRL in the Ant environment is not satisfactory, with unignorable reward gap compared to TD7  and other baselines (Fig. 5). Although the action smoothness of PASRL is better than other methods, the sacrifice on reward performance is unacceptable in application scenarios.
   2. The experiment is only conducted in the Ant task. It is necessary to evaluate the performance of each method in more environments, such as Hopper, HalfCheetah, Humanoid, etc.
   3. More baseline methods are necessary to demonstrate the performance of PASRL. As described above, some DRL methods have been proposed to improve action smoothness, such as LipsNet, TAAC, and some regularization methods introduced in the paper (SR^2L[4], L2C2[5], etc.). Please conduct experiments with additional baseline methods to demonstrate the superior performance of PASRL compared to these existing methods.

3. The expression of this paper needs to be revised for readiability,:

   1. Add punctuations, especially commas in some sentences. This is significant to remove ambiguity and  improve readiability.
   2. Some notations and terms are used without clear definitions. For example, in line 214 section 4.2, the readers cannot find the definition of $nh_t$. Some terms including LPF and ERE are used without claiming their full names. Please revise the manuscript accordingly.
   3. Some latex typos, such as ``representational drift'' in Line 105.

[1] LipsNet: A Smooth and Robust Neural Network with Adaptive Lipschitz Constant for High Accuracy Optimal Control

[2] TAAC: Temporally Abstract Actor-Critic for Continuous Control

[3] Addressing Action Oscillations through Learning Policy Inertia

[4] Deep Reinforcement Learning with Robust and Smooth Policy

[5] L2C2: Locally Lipschitz Continuous Constraint towards Stable and Smooth Reinforcement Learning

**Questions:**

1. As shown in Fig.5,  the action smoothness results of each method during training are given. Are these results evaluated in training mode (the agent takes actions with exploration noises) or in evaluation mode (without exploration noises)?
   Please clarify this issue in the paper, which has a significant influence on the experiment result.

---

> ### Author Response · Authors · 2024-11-19
>
> We would like to thank the reviewer for their quality review and suggestions to improve the quality of this work.
>
> Please find the detailed responses to your comments below.
>
> •Weaknesses 1: More related works need to be introduced in this paper. Improving action smoothness is the key contribution of this paper, which has been widely researched recently in DRL domain. The following are some related works:
>
> Response: Thank you for recommending these additional papers, for our literature. Please find them included in the revised manuscript’s introduction section.
>
> •Weakness 2: The experiment is insufficient and cannot demonstrate the effectiveness of the new method:
>
> Response: The point of the experiments is not to achieve a state of the art results, however to convey that our method is able to find a sweet spot between rewards and action smoothness, with reduced complexity compared to RL and traditional methods. While we do agree that the gap between TD7 and PASRL is apparent, only RL based controllers are not implemented into real world applications due to multiple reasons such as issues with action smoothness. Regarding additional baselines please find the revised manuscript with further additional baselines such as Hopper, Humanoid and Walker2D. Regarding the baselines comparing PASRL and LipsNet and TAAC we agree that it would prove insightful, but due to time constraints it is not included in the revised manuscript.
>
> •Weakness3: The expression of this paper needs to be revised for readiability,:
>
> Response: We have revised the script with additional care for punctuality and readability. Also we added definitions for nht, LPF and ERE.
>
> •Question 1: As shown in Fig.5, the action smoothness results of each method during training are given. Are these results evaluated in training mode (the agent takes actions with exploration noises) or in evaluation mode (without exploration noises)? Please clarify this issue in the paper, which has a significant influence on the experiment result.
>
> Response: The results were evaluated in evaluation mode (without exploration noise). We used different fixed seeds for the experiments. All baselines and our method were evaluated on the same fixed seeds as mentioned in the revised manuscript. The evaluation was performed every 5000 time steps, with checkpointing enabled for TD7. The results were averaged over the 10 evaluation episodes inside a seed and then were averaged across seeds. Windowing smoothing with a window size of 10 was applied to display the results. We have added this as a reproducibility statement in the appendix.

---

> > ### Comment · Reviewer_sexv · 2024-11-26
> > **Thanks for your response**
> >
> > We thank the authors for their efforts to improve the work and respond to the concerns. However, there still exist some key issues:
> > 1. the authors claim that "our method is able to find a sweet spot between rewards and action smoothness". I agree that RL agents need to take balance between reward performance and smoothness, which is commonly needed in existing works. However, how to guarantee the sweet spot?  This is not described in methodology or demonstrated in the experiment.
> > 2. As shown in Table 1&2, the new method achieves unignorable performance costs compared to TD7 with smoothness improvement. Can this method adjust the balance through tuning a hyper-parameter manually/automatically? This is quite important for the method in practical applications.
> > 3. The experiments are still insufficient to demonstrate the effectiveness. Some necessary baseline methods are absent, such as previous smooth RL methods. Besides, the results of the new method seem unsatisfactory. In some envs such as Walker2d, PASRL obtains lower performance in both rewards and smoothness, compared to TD7+AC+LPF+PD.
> >
> > Above all, there still exist some concerns in this paper. Therefore, I maintain the current rating.

---

> > > ### Author Response · Authors · 2024-11-29
> > >
> > > We thank the reviewer for their answer and engagement in the discussion. We acknowledge these limitations and issues of the current implementation.
> > >
> > > Regarding question 1, we cannot guarantee it theoretically, we have preliminary empirical results for the TD7 implementations action smoothness surface and PASRL’s action smoothness surface. Since it’s preliminary nature we have decided to not include this in the main text.
> > >
> > > Response 2: The method does not have a hyperparameter to adjust the performance/smoothness by itself. The smoothness is the byproduct of using time dependent embeddings.
> > >
> > > Response 3: We agree that, the results need improvement.
> > >
> > > Thank you once again for raising your valid concerns and questions to improve the paper.

---

### Official Review · Reviewer_RQLj · 2024-10-30

**Soundness:** 2
**Presentation:** 3
**Contribution:** 2
**Rating:** 3
**Confidence:** 5

**Summary:**

This paper proposed a mathod, PASRL, as an extension toTD7 which address instability and oscillations in the smoothness of concurrent actions.

**Strengths:**

1) PASRL introduce an approach to enhance action smoothness by incorporating past state-action representations through a RNN.
2) PASRL use pink noise  rather than standard white noise.

**Weaknesses:**

1) the main weakness is the optimality issue. Action plays a significant role in RL (exploration). Ideally, in order to reach optimal Q value ($Q^*(s,a)$) and optimal policy ($\pi^*$), you need to visite all state action pair (s,a) since  ($Q^{\pi^*}(s,a) \geq Q^\pi(s,a)$). During training, when you smooth the action, you will take the risk of limited exploration which affect the optimality, especially in high-dimensional environments such as humanoid.
2) Limitation: smooth action is not always best. Consider a well-known control example, racing car, the objective is only minimizing time. In this case bang-bang control is the optimal solution however it is not smooth.
3) Simulation examples are very limited only ant and hopper.

**Questions:**

1) Auther memtioned "The reliance on ONLY the current time step information ... causes instability and oscillations", for off-policy method such as TD3 and TD7, the data comes from reply buffer can be past time step information. Can you make the point more clear?
2) For Figure 7, how can the plot has shaded area for single seed line? when you do the experiment, I assume you change the environment seed. Do you fix the random seed, torch seed and/or numpy seed?
3) For Figure 5, the red, blue, and brown line, espicially red line has lowest reward but result in the best action smoothness. Can the auther explan why the better action smoothness will hurt the reward performance?

**Details Of Ethics Concerns:**

Simulation based paper, no need to have ethic review

---

> ### Author Response · Authors · 2024-11-19
>
> We thank the reviewer for their insights and valid concerns regarding the manuscript.
>
> We aimed to adress al the reviewer's concerns in the revised manuscript.
>
> Please find the individual answers to your questions below.
>
> •Weakness 1: the main weakness is the optimality issue. Action plays a significant role in RL (exploration). Ideally, in order to reach optimal Q value (Q∗(s,a)) and optimal policy (π∗), you need to visite all state action pair (s,a) since (Qπ∗(s,a)≥Qπ(s,a)). During training, when you smooth the action, you will take the risk of limited exploration which affect the optimality, especially in high-dimensional environments such as humanoid.
>
> Response: Yes, you are absolutely correct about the fact that the smoothing of the actions can limit exploration. However in our method we do not smooth actions, as we only aim to learn representations with time dependence, which have the effect of causing smoother actions. We have also included this caveat into our introduction section when discussing the importance of action smoothness. Furthermore we can attest to your comment by showing that the low-pass filter based hybrid methods can suffer from this exact limited exploration problem you mentioned.
>
> •Weakness 2:  Limitation: smooth action is not always best. Consider a well-known control example, racing car, the objective is only minimizing time. In this case bang-bang control is the optimal solution however it is not smooth.
>
> Response: Thank you for providing this example. Yes, you are right and we included this point in our revised manuscript version as a caveat in the introduction section. Our method aims to avoid jerky/twitching motion present in actuators/control methods, which even in the racing car example would prove important for the driver’s safety. Also we provide more evaluations to clarify that our algorithm does not oversmooth actions and it can still have highly responsive actions.
>
> •Weakness 3: Simulation examples are very limited only ant and hopper.
>
> Response: Please find additional baselines (Hopper, Humanoid and Walker2d) in the revised manuscripts evaluation section. Furthermore we also updated the “Effects of Representational drift” section, with additional environments (Humanoid, Hopper).
>
>
>
> •Question 1:Auther memtioned "The reliance on ONLY the current time step information ... causes instability and oscillations", for off-policy method such as TD3 and TD7, the data comes from reply buffer can be past time step information. Can you make the point more clear?
>
> Response: By only current time step information we meant that only a given timestep’s observation vector is present, regardless of in which timestep it was observed in.  We modified the text accordingly to make it more apparent to the readers. We believe that this limits the agent’s methods to learn information about the movement, since only a snapshot of given sensory information/ joint information is present without additional time dependent context.
>
> •Question 2: For Figure 7, how can the plot has shaded area for single seed line? when you do the experiment, I assume you change the environment seed. Do you fix the random seed, torch seed and/or numpy seed?
>
> Response: It is because we have evaluated each seed run for 10 evaluation episodes, and the shaded area shows the standard deviation in the given evaluation episodes. Yes, we fix them and we performed experiments with seed values mentioned in the manuscript.
>
> •Question 3: For Figure 5, the red, blue, and brown line, espicially red line has lowest reward but result in the best action smoothness. Can the auther explan why the better action smoothness will hurt the reward performance?
>
> Response: In that case the better action smoothness is not what hurts the reward performance, but the use of a low-pass filter (LPF) on top of a reinforcement learning (RL), agent. This is due to the systems being optimized independently from one-another and is mentioned in the introduction: “However what these methods do not take into account is that these modifications alter the original system’s closed-loop dynamics system leading to erratic control behavior”. As well as we would refer back to question 1: where the reviewer mentioned limited exploration by the smoothing of the actions during training, this is exactly the case here. Furthermore, we elaborate on this finding more in depth in the revised version’s result’s section.

---

### Official Review · Reviewer_KaSH · 2024-11-03

**Soundness:** 3
**Presentation:** 3
**Contribution:** 1
**Rating:** 3
**Confidence:** 4

**Summary:**

The paper tackles the problem of action smoothness, which is especially crucial for robotic systems that are incapable of instant changes of the motor torques.
They propose augmenting the state with the prior actions, and perform an ablation to showcase the performance differences.

**Strengths:**

The problem itself is important. When training for real world deployment, the constraints of the real systems need to be taken into account.

**Weaknesses:**

I'm not sure what the authors try to convey in terms of success/failure.
Their method is better at predicting smoother actions, this is not surprising. But also not surprising is the degradation in performance. This is a result of over-regularization.

Real systems don't necessarily need the "smoothest" policy possible, but rather have some constraints as to how reactive they can be.
I think this work should be extended to the constrained RL setting. Providing the required smoothness parameter as a part of the optimization process.

Then, the authors can compare with prior methods with differing hyper-parameters.
In the constrained RL setting, you want to show that your work is (1) easier to tune or requires no tuning, whereas other works need delicate reward design. (2) for a given constraint, out of all methods that meet the constraints, your controller reaches the best performance on the task itself.

**Questions:**

-

---

> ### Author Response · Authors · 2024-11-19
>
> Firstly we would like to thank the reviewer for taking the time to give a detailed and insightful review of our work.
>
> All the comments have been taken into account to improve the quality of the manuscript.
>
> Please find below the detailed responses to your comments:
>
> •Weakness 1: I'm not sure what the authors try to convey in terms of success/failure. Their method is better at predicting smoother actions, this is not surprising. But also not surprising is the degradation in performance. This is a result of over-regularization.
>
> Response: Our aim was to create a solely RL based control method which serves as a solution to commonly found RL + traditional control based controllers, which cannot be optimized as a closed loop dynamic system. We have updated the abstract and conclusions to emphasize this. Could you please elaborate on what you mean by over-regularization in this case?
>
> •Weakness 2: Real systems don't necessarily need the "smoothest" policy possible, but rather have some constraints as to how reactive they can be. I think this work should be extended to the constrained RL setting. Providing the required smoothness parameter as a part of the optimization process.
>
> Response: We agree that real systems do not need the “smoothest” policy possible, but we think it is a relevant problem which limits the applications of RL based controllers. In our evaluations we extend this work to constrained RL setting where we compare our method to a TD7 based agent, which is trained in common Gymnasium environments with added smoothness reward part, therefore optimizing its performance for action smoothness as well.
>
> •Weakness 3: Then, the authors can compare with prior methods with differing hyper-parameters. In the constrained RL setting, you want to show that your work is (1) easier to tune or requires no tuning, whereas other works need delicate reward design. (2) for a given constraint, out of all methods that meet the constraints, your controller reaches the best performance on the task itself.
>
> Response: Our aim is to show that PASRL is easier to tune compared to regularly found RL+traditional based controllers and environments where the reward function also optimizes for action smoothness.

---

### Meta-Review · Area_Chair_ST4G · 2024-12-17

**Metareview:**

This paper proposes PASRL, a deep reinforcement learning method aiming to achieve smoother actions in continuous control tasks, which is crucial for real-world robotic systems. It extends the TD7 algorithm by incorporating past action-state representations through a recurrent neural network and utilizing pink noise for exploration. While the paper tackles an important problem with a novel approach, Reviewers raise concerns about its limited contribution and potential negative impact on exploration and optimality. The idea of simply adding past actions to the state representation is considered somewhat incremental, and excessive smoothing might hinder finding the best policy. Furthermore, the evaluation is limited to two simulated environments, and comparisons with other action smoothing techniques are missing.  Reviewers also point out issues with clarity and presentation.

Overall, the paper needs further development to convincingly demonstrate its effectiveness and address concerns regarding novelty, exploration, and the thoroughness of evaluation.

**Additional Comments On Reviewer Discussion:**

The discussion phase has been focused mostly on the significance of the contribution and on the empirical evaluation. The Authors' rebuttal has not been sufficient to change the initial negative Reviewers' assessment.

---

### Decision · Program_Chairs · 2025-01-22

Reject